# Endoscopic Ultrasound Guided Fine Needle Aspiration versus Endoscopic Ultrasound Guided Fine Needle Biopsy for Pancreatic Cancer Diagnosis: A Systematic Review and Meta-Analysis

**DOI:** 10.3390/diagnostics12122951

**Published:** 2022-11-25

**Authors:** Galab M. Hassan, Louise Laporte, Sarto C. Paquin, Charles Menard, Anand V. Sahai, Benoît Mâsse, Helen Trottier

**Affiliations:** 1Division of Gastroenterology, Department of Medicine, Réseau Hospitalier Neuchâtelois, 2000 Neuchâtel, Switzerland; 2School of Public Health, Université de Montréal, Montreal, QC H3T 1C5, Canada; 3Research Center, CHU Sainte-Justine, Montreal, QC H3T 1C5, Canada; 4Department of Gastroenterology, Centre Hospitalier de l’Université de Montréal, Montreal, QC H2X 3E4, Canada; 5Department of Gastroenterology, Centre Hospitalier Universitaire de Sherbrooke, Sherbrooke, QC J1H 4C6, Canada

**Keywords:** fine-needle aspiration, fine-needle biopsy, pancreatic cancer, diagnostic accuracy

## Abstract

Introduction: One of the most effective diagnostic tools for pancreatic cancer is endoscopic ultrasound-guided fine-needle aspiration (EUS-FNA) or biopsy (EUS-FNB). Several randomized clinical trials have compared different EUS tissue sampling needles for the diagnosis of pancreatic cancer. Objective: To compare the diagnostic accuracy of EUS-guided FNA as EUS-FNB needles for the diagnosis of pancreatic cancer using a systematic review and meta-analysis. Method: A literature review with a meta-analysis was performed according to the PRISMA guide. The databases of PubMed, Cochrane and Google Scholar were used, including studies published between 2011–2021 comparing the diagnostic yield (diagnostic accuracy or probability of positivity, sensitivity, specificity, predictive value) of EUS-FNA and EUS-FNB for the diagnosis of pancreatic cancer. The primary outcome was diagnostic accuracy. Random effect models allowed estimation of the pooled odds ratio with a confidence interval (CI) of 95%. Results: Nine randomized control trials were selected out of 5802 articles identified. Among these, five studies found no statistically significant difference between the EUS-FNA and EUS-FNB, whereas the other four did. The meta-analysis found EUS-FNB accuracy superior to EUS-FNA for the diagnosis of pancreatic cancer with a pooled odds ratio of 1.87 (IC 95%: 1.33–2.63). Conclusion: As compared to EUS-FNA, EUS-FNB seems to improve diagnostic accuracy when applied to suspicious pancreatic lesions.

## 1. Introduction

Pancreatic cancer is one of the most deadly cancers in the world [1]. In 2008, more than 400,000 people developed a pancreatic adenocarcinoma and as many people have died from this pathology [2]. Its incidence varies by country. For example, in China in 2018, the annual cumulative impact of pancreatic cancer was evaluated at 201.7/100,000, and on the other hand, in the United States and United Kingdom, it was, respectively, about 352.2/100,000 and 319.2/100,000 [3]. Pancreatic cancer represents a major issue in oncology because its prognosis remains poor with net survival rate at 5 years under 10% [4]. Early and precise detection improve prognosis. The best diagnostic tool is EUS with EUS-guided biopsy of suspicious lesions [5]. There are several types of needles for EUS-guided tissue sampling: standard cytology needles, and needles designed to produce histological cores. Core needles can be subdivided into “scraping”-type (forward and reverse bevel tip design) and “cutting”-type (Franseen tip and shark tip). Diagnostic accuracy is the primary outcome in most studies comparing the yield of different needle types. The diagnostic accuracy of FNA needles for pancreatic cancer is generally lower than for FNB needles [6,7,8,9]. The aim of this systematic review and meta-analysis was to compare the diagnostic yield of EUS-FNA cytology needles with that of the various EUS-FNB needles.

## 2. Methods

The literature review and the meta-analysis were performed according to the PRISMA guide.

### 2.1. Information Source and Search Strategy

The following databases were used: PubMed, Cochrane and Google Scholar. The search allowed us to select all studies published between 1 January 2011 and 12 May 2021. We also reviewed the list of references of the articles retained.

### 2.2. Keywords—Search Equation

These hierarchical keywords Medical Subject Headings were used: (endoscopic ultrasound fine needle aspiration or standard fine needle endoscopy ultrasound) and (endoscopic ultrasound fine needle biopsy OR new fine needle ultrasound endoscopy) AND (pancreatic masses OR pancreatic cancer) AND (diagnosis OR diagnostic precision).

### 2.3. Eligibility Criteria, Articles Selection

To be eligible, the retained studies had to be randomized, controlled trials published in English, having compared the diagnostic yield of EUS-FNA and EUS-FNB of the solid pancreatic masses with a number of patients greater than or equal to 30. Systematic revues or meta-analyses were not included [10].

Two assessors (GM, LL) independently examined the title/summary and disagreements were resolved with co-authors (HT). The article selection was done in three steps. During the first step, all articles whose titles related to the diagnosis of pancreatic cancer by endoscopic ultrasound were enumerated by eliminating duplicates. Reading all selected article summaries allowed, in the second step, to conserve the potential articles. The last step consisted of a full reading of all articles included in the search and verification of all the inclusion criteria.

### 2.4. Data Extraction and Quality Analysis of the Retained Studies

The data were extracted from retained studies by using a pre-designed form that aimed to document the following elements: author, year, title, study design, population, intervention, diagnostic precision and other diagnostic measures with the main results (*p*-value or association measure with their confidence interval (95% CI)).

The quality of the studies was evaluated using the Cochrane Collaboration’s tool for assessing risk of bias in randomized trials [11] and the quality assessing tool of Health Evidence served as instruments to measure the quality score of the studies [12]. The grid consists of ten evaluated items to answer positively (corresponding to 1 point) or negatively (corresponding to 0 point). The score of 0 point is also attributed to an item if the information in the article isn’t allowed to be evaluated. The sum of the number of points obtained for each item is then counted to provide a global score that allows us to classify the studies according to the level of quality: score 8–10 (rigorous level); score 5–7 (average level) and score ≤ 4 (low level).

### 2.5. Data Synthesis

A pooled odds ratio for diagnostic accuracy was estimated by using the method of Mantel–Haenszel. A forest plot was created to show the observed odds ratio (OR) dispersion of the diagnostic precision using EUS-FNA as the referent compared to EUS-FNB. The I2 statistic with its 95% CI, was used to quantify the proportion of variance in the observed OR reflecting the heterogeneity between the studies. Statistical analyses were performed the Review Manager computer program (RevMan software version: 3.2.0, United States).

## 3. Results

### 3.1. Selected Articles

A total of 5802 articles were identified, of which 5793 were eliminated, based on the previously described selection algorithm (Figure 1). Nine articles were retained for analysis.

### 3.2. Study Characteristics

The characteristics of the studies are presented in Table 1. In all nine articles, one study was a double blind randomized clinical trial [13] and the others were single blind, randomized clinical trials [14,15,16,17,18,19,20,21]. The studies were performed in the United States (n = 3), in the United Kingdom (n = 2), China (n = 2) and South Korea (n = 2). The number of patients included in the studies varied from 36 to 1088 patients.

All studies included the main outcome, diagnostic accuracy [13,14,15,16,17,18,19,20,21]. Some authors did not find a statistically significant difference between the diagnostic accuracy of the two needle types tested [13,14,15,17,18]; whereas others found that the EUS-FNB needle showed statistically significantly better accuracy than the EUS-FNA needle tested [16,19,20,21].

Other outcomes of the studies were varied. Two studies had also assessed sensitivity, specificity and predictive values [13,19]. According to Tian et al. (2018), the negative predictive values, the sensitivity and specificity were, respectively, 50%, 80% and 100% for the EUS-FNB and 82%, 78% and 100% for the EUS-FNA (*p* > 0.05), while the positive predictive value was 100% for both methods [13]. However, Naweed et al. (2018) found that the differences were not statistically significant for the sensitivity and specificity between the two groups [19,20,21,22,23].

Two studies looked at the sampling duration and the time added for rapid onsite pathological evaluation (ROSE) [20,21,22,24,25,26,27]. In the study of Oppong et al. (2020), the median time for diagnosis was about 188 s for EUS-FNB vs. 332 s for EUS-FNA (*p* < 0.001) [21,27,28].

Other studies analyzed the number of needle passes required [15,17,19]. As a whole, these studies agreed that the EUS-FNB seems to require less needle passes, with reduced time to diagnosis. Tian et al., found that the EUS-FNA required more passes compared to EUS-FNB (1.83 vs. 1.11, *p* < 0.05) [13]. Similarly, Naveed et al. found the median number of passes needed for diagnosis with the EUS-FNB was significantly lower than that with EUS-FNA (1 vs. 3, *p* < 0.001) [19], as did Lee et al., (1 vs. 2, *p* < 0.001) [17]. Only Bang et al. found no difference in the median number of passes required [15].

Some authors evaluated the histological sample quality, and mostly found EUS-FNB provides higher quality samples than EUS-FNA [14,15,16,21]. EUS-FNB sample quality was superior to EUS-FNA: Cheng et al. (91.44% vs. 80.0%, *p* = 0.0015) [16], Aadam et al. (90.0% vs. 61.7%, *p* = 0.002) [14], Bang et al. (80% vs. 66.7% *p* = 0.06) [15].

Finally, Oppong et al. (2020) have also evaluated the diagnostic facility and the cost of both of the diagnostic methods [21]. They observed that practicing the EUS-FNB was associated with a higher diagnostic facility when there was any significant difference in terms of financial costs against the types of needles [23,24,25,26].

### 3.3. Study Quality Analysis

Figure 2 presents the results of quality analysis of the retained studies. Five studies [13,15,16,18,21] have a score higher or equal to 8, implying only a small risk of bias. But another four studies have a score quality ranging from 5–7, meaning they had an average risk of bias.

### 3.4. Quantitative Synthesis of the Results

The meta-analysis results are presented in Figure 3 and show that the EUS-FNB has greater diagnostic accuracy than EUS-FNA for suspicious pancreatic lesions, with a pooled OR (95% CI) of 1.87 (1.33–2.63) for EUS-FNB compared to EUS-FNA.

## 4. Discussion

The results of this meta-analysis show that the EUS-FNB provides better diagnostic accuracy than EUS-FNA for suspicious pancreatic lesions (pooled OR (95% CI) = 1.87 (1.33–2.63). As compared to previous meta-analyses, the strength of ours lies in that we included only controlled and randomized studies with a near null rate of heterogeneity and that focused on sampling of suspicious pancreatic masses only. On the other hand, the number of included studies (n = 9) remains low; and, as usual, publication bias in favor of positive studies, and against studies with results supporting the null hypothesis could reduce the external validity of our conclusions.

Wang et al., compared EUS-FNA to EUS-FNB and found a pooled OR (95% CI) of 0.72 (0.49–1.07) for the diagnostic accuracy. This meta-analysis had the same inclusion criteria as ours, but included fewer studies; which could explain why they found no statistically significant difference between both needle types. 

A more recent and more widely inclusive meta-analysis (including cancerous and non-cancerous lesions) by Renelus et al. also found EUS-FNB was superior to EUS-FNA (87% and 81%, *p* = 0.005). It included 12 randomized controlled trials published between 2012 and 2019 [7]. Similarly, Van Rietet al. included 14 randomized controlled trials on cancerous and non-cancerous lesions and found that the EUS-FNB was superior to EUS-FNA (diagnostic accuracy 87% vs. 80%, *p* = 0.002) [8]. Finally, a network meta-analysis by deHan et al., including cancerous and non-cancerous lesions also showed that EUS-FNB was superior to EUS-FNA [9].

Unfortunately, subgroup analysis study of more subtle differences in the accuracy of various needle tip types and diameters was limited due to inadequate sample sizes. For example, there are data suggesting that newer “cutting” tip needles are superior to “scraping” tip needles [21,22,23]. In sensitivity analysis, we pooled all the available data on these two needle types, to compare their diagnostic accuracy. We found a non-statistically significant higher diagnostic accuracy for the cutting needles, with a combined OR (95% CI) 1.47 (0.67–3.22).

Generally, the literature suggests an improvement of the diagnostic precision obtained by the EUS-FNB needles in case of pancreatic cancers but, also, for the whole of solid pancreatic lesions. This amelioration could be explained by the fact that these last are designed toward obtaining better tissues samples with a better quality while conserving the histological characteristics of the tissues. However, it must be noted that many factors can affect the diagnostic precision, such as the location and the characteristics of the lesion, the technician’s experience, the needles caliber and the pass amount [22,23]. For example, Jani et al. (2019) in their study, observed that optimal attainment of tissues from the lesions depends on factors like the size of the needles caliber, the presence of cytotechnicians for a speed evaluation on sight, the valuation of the endoscopist and the manipulation techniques of the tissues [25,26,27,28]. Future research could look at the impact of those other factors on the EUS-FNB diagnostic yield [29,30].

Our study has strengths and limitations. The quality of our work relies on the fact that we only included controlled and randomized studies about cancerous pancreatic masses that showed a rate of heterogeneity near of null. On the other side, the number of included studies (n = 9) remains small. It would also be relevant to make an analysis within the diverse needles caliber existing but their number did not allow subgroup analysis. Besides, the results of our study could be marred by the bias of publication because studies are rarely published with results supporting the null hypothesis.

## 5. Conclusions

Pancreatic cancer remains to be a specific condition of concern with a high morbidity and a reduced survival, mainly when the diagnosis is tardily made. Then, it is imperious to improve the diagnostic means for a precocious detection. As compared to EUS-FNA, EUS-FNB seems to improve diagnostic accuracy when applied to suspicious pancreatic lesions.

Obtaining diagnostic precision with the fewer needles passage is still a predilection argument for the EUS-FNB needles in case of solid pancreatic masses. However, some factors can have an impact on the diagnostic accuracy such as: the type of the puncture needles, the puncture track, and the number of passage, the size of the tumor and the endoscopist’s experience of the pathology. Future studies are needed in order to watch the effect of other factors on the diagnostic yield.

## Figures and Tables

**Figure 1 diagnostics-12-02951-f001:**
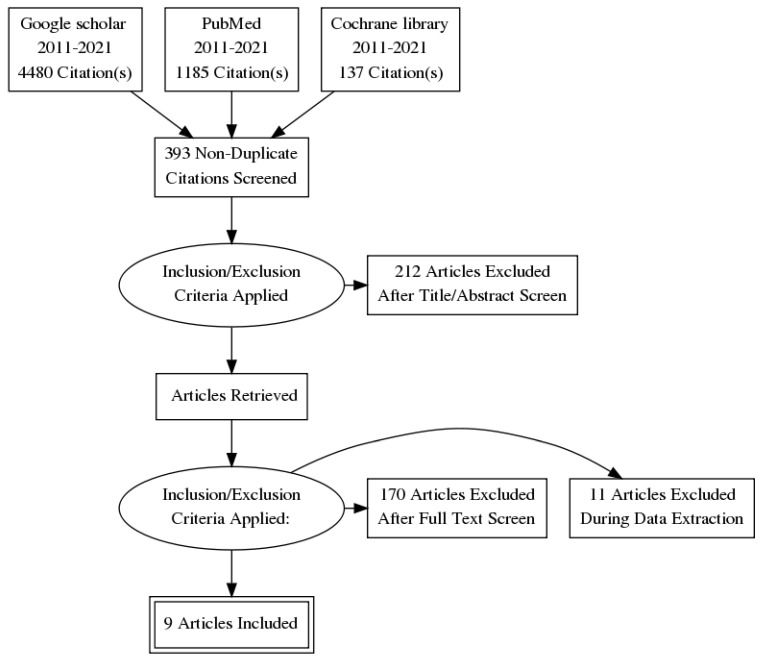
Studies selection diagram. Following text reading, 170 articles was excluded for the following reasons: other lesions not pancreatic, studies other than randomized clinical trial or insufficient data (inaccurate study population such as abdominal solid mass, sample size less than 30 patients, no data available on the diagnostic yield). After this step, 11 articles have been removed during the extraction phase of data, because they did not give the measurement of the diagnostic yield about sampling techniques in case of cancerous pancreatic lesions (only for the gathered solid pancreatic lesions involving, for example, pancreatic cysts and chronic pancreatitis).

**Figure 2 diagnostics-12-02951-f002:**
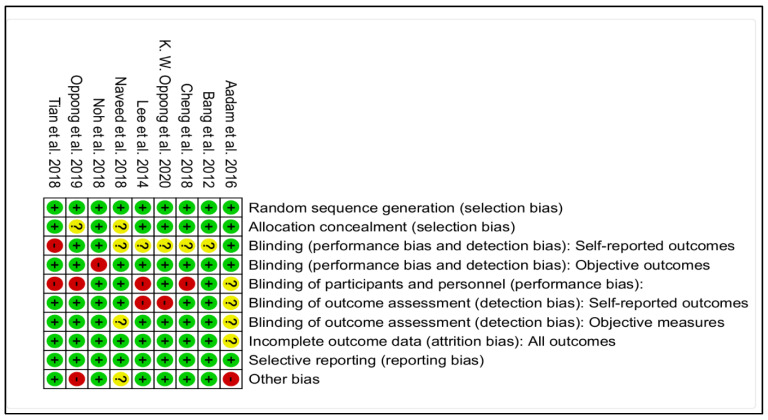
Evaluation of studies quality. Items evaluated as positive for the quality (in green); Negatively evaluated items (in red); The available information in the articles did not allow the evaluation for this item (in yellow) [13,14,15,16,17,18,19,20,21].

**Figure 3 diagnostics-12-02951-f003:**
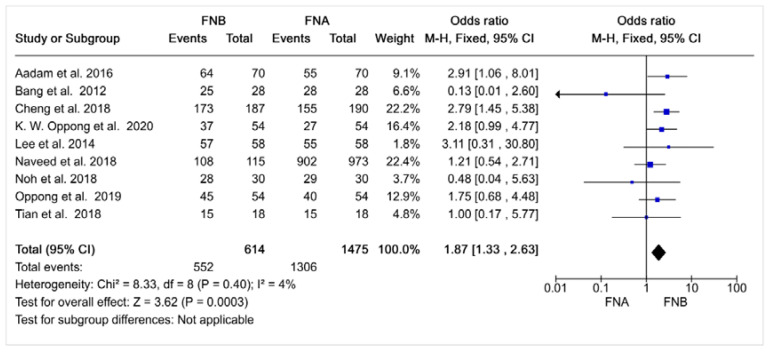
Forest plot for diagnostic precision. I^2^: Heterogeneity rate; M-H: Odds ratio according to the method of Mantel–Haenszel; CI = Confidence Interval; Events = positive cases; FNA: Fine Needle Aspiration; FNB: Fine Needle Biopsy [13,14,15,16,17,18,19,20,21].

**Table 1 diagnostics-12-02951-t001:** Retained studies characteristics.

Authors	Title	Type	Country, Patients and Lesions Type	Method and Needle Type	Diagnostic Precision	Other Results	Conclusion
Tian et al. (2018) [13]	*«Evaluation of 22G fine-needle aspiration (FNA) versus 22G fine-needle biopsy (FNB) for endoscopic ultrasound-guided sampling of pancreatic lesions: a prospective comparison study»*	Prospective comparative studies	Studies realized in China on 23 men and 13 women at average age of 59.5 ± 19.5 years old (45–75 years old) with solid pancreatic masses. Total number of included patients: 46.	Patients were randomly allocated in two groups: EUS-FNA (n = 18) and EUS-FNB (n = 18) Using 22G EUS-FNB needles (EchoTipProCore, Cook Medical) and 22G EUS-FNA (Olympus, GF UCT 160).	The precision of the diagnostic was about 83.3% for both techniques (*p* > 0.05).	The negative predictive value, the sensitivity and the specificity were, respectively, 50%, 80% and 100% for EUS-FNB and 82%, 78% and 100% for EUS-FNA. The positive predictive value was about 100% for both of the techniques. The EUS-FNB needed less punctures than the EUS-FNA (1.11 vs. 1.83 *p* < 0.05).	22G EUS-FNB is a sure and effective way to diagnose the solid pancreatic masses and the EUS-FNB needs a lower number of needle pass to establish a diagnostic compared with EUS-FNA
Oppong et al. (2019) [20]	«*PWE-072 EUS Fork-tip biopsy versus EUS FNA in the diagnosis of solid pancreatic masses*»	Randomized controlled trial	Studies performed in the United Kingdom on 108 patients with 57 men at an average age of 66.9 ± 10.9 years old with solid pancreatic masses. Total number of included patients: 108.	Each patient went through three passes with a EUS-FNA needle (Beacon 25G and 22G) and three passes with a cored EUS-FNA (SharkCore 25G and 22G) randomly made. 25G needles were used for trans-duodenal sampling and 22G for trans-gastric.	The diagnostic precision of the EUS-FNB was significantly superior to that of the FNB (84.2% vs. 75% *p* = 0.041).	The notification time of the pathology (191 s vs. 332 s *p* < 0.0001) was significantly shorter with the EUS-FNB than with the EUS-FNA EUS-FNB had a diagnostic tool more abundant (59.2% vs. 44.4% *p* = 0.017) and an easier diagnostic (68.9 % vs. 51.9% *p* = 0.03) than EUS-FNA	SharkCore Needle were significantly better than the EUS-FNA standard needle in diagnosing the solid pancreatic masses and associated to a better quality of sample, a facility of time notification of sampling and a shorter pathology notification
Naveed et al. (2018) [19]	«*A multicenter comparative trial of a novel EUS-guided core biopsy needle (SharkCore™) with the 22-gauge needle in patients with solid pancreatic mass lesions*»	Multicenter retrospective comparative studies	Studies made in the United States of 1088 patients where 533 were women. The average age of the patients was 66 years old with solid pancreatic masses. Total number of included patients: 1088.	115 sustained a EUS-FNB with a 22G SharkCore needle and 973 sustained standard EUS-FNA of 22G (EchoTip Ultra 3 needle; Wilson-Cook Medical, Winston-Salem, North Carolina)	The diagnostic precision was about 94.1% for the EUS-FNB and 92.7% for EUS-FNA *p* = 0.85	The difference was not statistically significant for the sensitivity and the specificity in the two groups. The median number of passes to obtain a tissue diagnostic by using EUS-FNB was significantly inferior to that of the standard needle (1 vs. 3 *p* < 0.001).	The EUS-FNB has a diagnostic yield similar to the EUS-FNA standard needle and greatly reduces the amount of necessary needle passes to get the tissues diagnostic.
Cheng et al. (2018) [16]	«*Analysis of Fine-Needle Biopsy* vs. *Fine-Needle Aspiration in Diagnosis of Pancreatic and Abdominal Masses: A Prospective, Multicenter, Randomized Controlled Trial*»	Prospective trial, multicenter, controlled and randomized	Studies made in China with 377 patients with solid pancreatic masses with 232 men. The average age was 58 years old (249 pancreatic masses). Total number of included patients: 377.	Patients were randomly allocated into groups for an evaluation 22G EUS-FNA (n = 190 and 22G EUS-FNB (n = 187). Group A, using commercially available FNA needles (22G EchoTip Ultra needle, Cook Medical); Group B, using the EUS-FNB needles (22G EchoTip ProCore needle, Cook Medical).	The diagnostic precision was about 92.68% for EUS-FNB vs. 81.75% for EUS-FNA (*p* = 0.0099) while the cytological precision was about 88.62% (EUS-FNB) vs. 79.37% (EUS-FNA) (*p* = 0.00468).	The sampling for the EUS-FNB was about 91.44% vs. 80% for the EUS-FNA (*p* = 0.0015).	The samples obtained through EUS-FNB needles produced a more accurate diagnostic than the samples collected with EUS-FNA needles for the pancreatic masses.
Bang et al. (2012) [15]	*«Randomized trial comparing the 22-gauge aspiration and 22-gauge biopsy needles for EUS-guided sampling of solid pancreatic mass lesions»*	Randomized controlled trial	Studies made in the United-States with 31 men and 25 women in the age between 57–77 with solid pancreatic masses. Total number of included patients: 56.	In total, 28 patients were randomly selected in the group of 22G EUS-FNB (Echotip ProCore; Cook Endoscopy, Bloomington, IN) and 28 in the group of 22G EUS-FNA (Expect; Boston Scientific, Natick, Mass).	The diagnostic precision was about 100% for the EUS-FNA and 89.3% for the EUS-FNB (*p* = 0.24).	No significant difference in the median number of needle passes for diagnosing pancreatic lesions. Noted 3.6% of complications for the two methods. Samples quality was about 80% for EUS-FNB vs. 66.7% for the EUS-FNA *p* = 0.66.	Lack of significant difference between the two needles for the diagnostic precision.
Aadam et al. (2016) [14]	*«A randomized controlled cross-over trial and cost analysis comparing endoscopic ultrasound fine needle aspiration and fine needle biopsy»*	Multicenter randomized crossed trial	Studies realized in the United States on 74 men and 66 women with an average age of 64 years old with solid pancreatic masses. Total number of included patients: 140	140 patients were involved and 70 of them were randomly divided in the group 22G and 25G EUS-FNA (Echotip^TM^, Cook Medical, Winston-Salem, NC; Expect^TM^, Boston Scientific, Natick MA) and the other 70 for 19G, 22G and 25G EUS-FNB (Echotip Procore^TM^, Cook Medical, Winston-Salem, NC).	The diagnostic precision was about 91.7% for the EUS-FNB vs. 78.4% for the EUS-FNA (*p* = 0.19).	The quality of the sample was better with the EUS-FNB (90%) vs. 67.1% for the EUS-FNA *p* = 0.002.	There is no statistical difference concerning the diagnostic performance between the 2 needles.
Oppong et al. (2020) [21]	«*Fork-tip needle biopsy versus fine-needle aspiration in endoscopic ultrasound-guided sampling of solid pancreatic masses: a randomized crossover study»*	Randomized crossed studies	Studies done in United Kingdom of 57 men and 51 women with an average age of 69 years old (30–87 years old) with solid pancreatic masses. Total number of included patients: 108	108 patients were recruited. 54 patients were divided in the 22G SharkCore needle group (EUS-FNB) and 54 in the 25G Beacon needle group (EUS-FNA)	The diagnostic precision was about 69% [IC 95% 60–78%] for the EUS-FNB and 51% [IC 95% 41–61%] for the EUS-FNA *p* < 0.001.	The median time of diagnostic observation was 188 s for the FNB vs. 332 s for the FNA (*p* < 0.001). There was a significant difference in the sensitivity: 82% [IC: 95% 72–89%] for the EUS-FNB and 71% [IC 95% 60–80%] pour l’EUS-FNA.	The diagnostic yields of FNB needle with forked tip were significantly better than that of FNA with a reduced time of pathology observation.
Lee et al. (2014) [17]	*«Core biopsy needle versus standard aspiration needle for endoscopic ultrasound-guided sampling of solid pancreatic masses: a randomized parallel-group study»*	Randomized controlled studies in parallel groups	Studies realized in South Korea of 116 patients with an average age of 63.1 years old for the FNA group and 66.7 years old for the FNB group, with a solid pancreatic masses. 61 men and 55 women. Total number of included patients: 116.	The patients with pancreatic masses were included in a prospective way and randomized, with 58 for the 22G EUS-FNB group (Echotip Procore^TM^, Cook Medical). and 58 for the 22G EUS-FNA (Echotip^TM^, Cook Medical)	The diagnostic precision was about 98.3% for the EUS-FNB and 94.8% for the EUS-FNA (*p* = 0.671).	The EUS-FNB needed a fewer number of median passes in comparison with the EUS-FNA (1 vs. 2; *p* < 0.001).	The diagnostic precision was the same for the 2 types of needles. However, there was less passes needed to establish the malignity diagnosis with the FNB.
Noh et al. (2017) [18]	*«Comparison of 22-gauge standard fine needle versus core biopsy needle for endoscopic ultrasound-guided sampling of suspected pancreatic cancer: a randomized crossover trial»*	Randomized crossed trial	Studies realized in South Korea of 60 patients aged between 18 and 80 years old with pancreatic masses. We can count 35 men and 25 women with an average age of 61.6 years old. Total number of included patients: 60.	A total of 60 patients with pancreatic cancers suspicion not resectable selected for a sampling guided by EUS were randomly allocated in two groups. 30 patients for 22G EUS-FNA (Olympus, Japan) and 30 patients for 22G EUS-FNB needles (EchotipProcore^TM^, Cook Medical, Ireland) realized in random order.	FNA and FNB needles reported respectively a level of diagnostic precision of 95% and 93.3% (*p* = 0.564).	The EUS-FNB showed a better quality of sample than the EUS-FNA.	The diagnostic precision of the sampling guided by EUS for pancreatic cancer by using 22G FNA was comparable to that of FNB needles.

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
