# Peer review of "Endoscopic Ultrasound Guided Fine Needle Aspiration versus Endoscopic Ultrasound Guided Fine Needle Biopsy for Pancreatic Cancer Diagnosis: A Systematic Review and Meta-Analysis"

_diagnostics, 2022, doi:10.3390/diagnostics12122951_

Round 1
Reviewer 1 Report
Endoscopic ultrasound guided fine needle aspiration versus endoscopic ultrasound guided fine needle biopsy for the pancreatic cancer diagnosis: A Systematic review and meta-analysis.
This is a well written e review articles of nine randomised clinical trials looking at the efficacy of US guided biopsy V FNA: Nine randomized control trials 5 of the 9 studies found no statistically significant difference between the EUS-FNA and EUS-FNB, whereas the other 4 did. The meta-analysis found, as expected, EUS-FNB accuracy superior to EUS-FNA for the diagnosis of pancreatic cancer . The authors concluded that EUS-FNB gives more diagnostic accuracy when applied to suspicious pancreatic lesions
Author Response
Many thanks to the reviewer for the comments. There are no specific questions.Once again, I thank the reviewer very much.
Reviewer 2 Report
Hassan and colleagues presented a systematic review and meta-analysis including 9 controlled and randomized studies comparing EUS-guided FNA and FNB in solid pancreatic lesions. The paper is well written and the results are similar to previously published articles.
There are some issues that need to be addressed.
1. In the table reporting and summarizing the included articles, data should be expressed more clearly. In detail, if the needle type for both, FNA and FNB, is not reported in the original paper please report that this information is not available. The same for the diameter of the needle.
2. In the Discussion the authors reported that only controlled and randomized studies were included, but in the Table, the article by Naveed et al. is classified as "multicenter RETROSPECTIVE comparative studies". Please clarify.
3. In the Table please report clearly the total number of included patients in each study (as a single number in a dedicated column).
4. About 50% of all included patients (1088) were extrapolated for the paper of Naveed et al. Might this give an additional bias to the paper. Please comment on this.
Author Response
Many thanks to the reviewer for the comments.
Below are the reviewer's questions and our answers follow each question:
1. In the table reporting and summarizing the included articles, data should be expressed more clearly. In detail, if the needle type for both, FNA and FNB, is not reported in the original paper please report that this information is not available. The same for the diameter of the needle.
Response: I have completed the table with the informations requested by the reviewer in points 1 and 3. You will also find this informations written in red in the table.
2. In the Discussion the authors reported that only controlled and randomized studies were included, but in the Table, the article by Naveed et al. is classified as "multicenter RETROSPECTIVE comparative studies". Please clarify.
Response: This study is effectively a comparative and retrospective study and not a prospective study. However, this is a study comparing the diagnostic accuracy between EUS-FNA and EUS-FNB. We limited ourselves to comparative studies and not only to prospective studies. The authors of this study also specify the limits of this retrospective study, which nevertheless has the merit of comprising a larger sample size.
3. In the Table please report clearly the total number of included patients in each study (as a single number in a dedicated column).
Response: I have completed the table with the informations requested by the reviewer in points 1 and 3. You will also find this informations written in red in the table.
4. About 50% of all included patients (1088) were extrapolated for the paper of Naveed et al. Might this give an additional bias to the paper. Please comment on this.
Response: This study, which is a retrospective study, compares the diagnostic accuracy, sensitivity, specificity, PPV and NPV between EUS-FNA and EUS-FNB with a study population of 1088 patients. The problem is that we have a gold standard (surgical histology) in only 327 patients and not for the rest. A verification bias is possible since it is only the positive cases that are operated on.

Reviewer 3 Report
First, thank you for giving me the opportunity to review this article. However, I have some concerns.
1. How about the results of meta-analysis for the number of aspirations, adverse events between EUS-FNA and FNB?
2. Cytology is enough for diagnosing pancreatic cancer. Sampling histological specimen by FNB needles is important for genome profiling. Could you mention that matter in introduction or discussion?
3. How about the difference of handiness between FNA needles and FNB needles?
Author Response
Many thanks to the reviewer for the comments.
Below are the reviewer's questions and our answers follow each question:
- How about the results of meta-analysis for the number of aspirations, adverse events between EUS-FNA and FNB? Response : the main research question of this meta-analysis was diagnostic accuracy and not adverse effects. however we already know that FNA is relatively safe with less than 3% adverse effects which are mainly represented by minor bleeding and/or pain. Also, we know that FNB generates more risk of bleeding.
- Cytology is enough for diagnosing pancreatic cancer. Sampling histological specimen by FNB needles is important for genome profiling. Could you mention that matter in introduction or discussion? Response: FNA which allows a cytological diagnosis of pancreatic cancer is far from being sufficient since it allows a diagnostic yield of 80 to 84% (3-10; 17-22) and above all a negative predictive value of only 60% (17-22).
This means that 4/10 the result of the cytology (FNA) is negative while it is pancreatic cancer. In clinical practice, in front of a patient suspected of pancreatic cancer, when the cytology is negative. The patient is taken again for a new FNA whose result is not guaranteed precisely because of this negative predictive value. Hence the development of technologies to optimize this result such as elastography, Rapid on Site evaluation (ROSE) or FNB needles. - How about the difference of handiness between FNA needles and FNB needles? Response: 25G needle (smaller diameter) are the needle with the best the handiness and 19G needle (larger diameter), whether FNA or FNB needles, are the most difficult to handle. Handiness is therefore more related to the diameter of the needle. The advantage of FNB needles is that we usually use 22G cutting needles like the Franseen to have the best diagnostic yield whereas for the FNA, we had to use the 19G which was much more difficult to handle.
Round 2
Reviewer 2 Report
The authors addressed the issues.
Reviewer 3 Report
None.